# DASH: Faster Shampoo via Batched Block Preconditioning and Efficient Inverse-Root Solvers

Ionut-Vlad Modoranu [1]   Philip Zmushko [1]   Erik Schultheis [1]   Mher Safaryan [2]   Dan Alistarh [1 3]

## Abstract

Shampoo is one of the leading approximate second-order optimizers: a variant of it has won the MLCommons AlgoPerf competition, and it has been shown to produce models with lower activation outliers that are easier to compress. Yet, applying Shampoo currently comes at the cost of significant computational slowdown, due to its expensive internal operations. In this paper, we take a significant step to address this shortcoming by proposing **DASH** (for **D**istributed **A**ccelerated **SH**ampoo), a faster implementation of Distributed Shampoo based on two main new techniques: First, we show that preconditioner blocks can be stacked into 3D tensors to significantly improve GPU utilization; second, we introduce the Newton-DB iteration and the Chebyshev polynomial approximations as novel and faster approaches for computing the inverse matrix roots required by Shampoo. Along with these algorithmic contributions, we provide a first in-depth analysis of how matrix scaling critically affects Shampoo convergence. On the practical side, our GPU-aware implementation achieves up to $5.6\times$ faster optimizer steps compared to the well-optimized Distributed Shampoo, while Newton-DB attains the lowest validation perplexity per iteration among all tested methods. Our code is available at https://github.com/IST-DASLab/DASH.

## 1. Introduction

The promise of faster adaptive gradient optimization methods has led to a full spectrum of methods inspired by the full-matrix AdaGrad optimizer (Duchi et al., 2011). At one end, the full-matrix approach offers theoretical guarantees but is computationally prohibitive for large models due to its $O(m^2n^2)$ memory complexity for an $m \times n$ layer. At the other end, diagonal approximations such as Adam (Kingma & Ba, 2014) and AdamW (Loshchilov & Hutter, 2017) reduce this complexity to $O(mn)$, and have become standard for deep learning. Yet, diagonal methods fail to capture complex parameter correlations, leading to significant work on efficiently incorporating non-diagonal information into optimizers, without incurring the prohibitive costs of full-matrix methods. One such instance is the Shampoo optimizer (Gupta et al., 2018; Anil et al., 2020), which captures layer-wise second-order information, while maintaining a manageable memory complexity of $O(m^2 + n^2)$, making higher-order optimization feasible for large-scale models.

Historically, Shampoo has remained in the shadow of standard diagonal optimizers like AdamW. Recently, however, Shampoo has gained significant traction, highlighted by its leading performance in the AlgoPerf benchmarking competition (Dahl et al., 2023), where it proved to be the best in terms of wall-clock time required to reach a target training performance. Further, recent studies suggest that converged solutions found by Shampoo possess desirable properties, such as improved generalization (Pascanu et al., 2025) and robustness to quantization (Vlassis et al., 2025).

While Shampoo's memory complexity is manageable, the optimizer still incurs a substantial computational overhead per optimizer step, relative to AdamW. Specifically, the algorithm requires computing inverse matrix roots, an operation that typically scales as $\Theta(n^3)$ for an $n \times n$ preconditioner matrix. To amortize this cost, standard implementations update the preconditioners infrequently (e.g., every 10–100 steps). However, this creates a trade-off between runtime speed and optimization quality: for example, evidence from Vyas et al. (2024) (see Figure 1 therein) indicates that more frequent preconditioner updates directly lead to better performance.

The Distributed Shampoo implementation (Shi et al., 2023) addresses some of these computational bottlenecks by splitting preconditioners into blocks of the size $B \times B$ with $B < n$, effectively reducing the complexity to $O(Bn^2)$. Yet, the underlying algorithms still have critical efficiency gaps: for instance, the default implementation still relies

[1]Institute of Science and Technology Austria [2]Lancaster University [3]Red Hat AI. Correspondence to: Ionut-Vlad Modoranu <ionut-vlad.modoranu@ista.ac.at>.

*Proceedings of the 43rd International Conference on Machine Learning*, Seoul, South Korea. PMLR 306, 2026. Copyright 2026 by the author(s).

on EVD[1], an operation that is notoriously difficult to parallelize on GPUs. Although Distributed Shampoo introduced a more GPU-friendly, matrix-multiplication-based alternative relying on the Coupled-Newton (CN) iteration, this is not enabled by default, likely due to concerns regarding numerical stability[2].

Motivated by this efficiency gap for Shampoo, in this paper we aim to significantly reduce the computational overhead of Shampoo while preserving its numerical precision. We focus on accelerating the inverse matrix root computation, which is the algorithm's primary bottleneck, by making high-quality preconditioning practical for widespread use. To this end, we propose **DASH** (**D**istributed **A**ccelerated **SH**ampoo), a high-performance implementation designed to fully leverage modern GPU architectures. By bridging the gap between theoretical efficiency and hardware support, **DASH** empowers researchers to investigate the optimizer's properties in diverse settings without prohibitive runtimes. Our contributions are as follows:

1. We introduce **DASH**, a novel GPU-efficient approach for block preconditioners. The sequential loops used in prior Distributed Shampoo implementation are replaced with independent blocks stacked into 3-D tensors and processed in parallel, to significantly increase GPU utilization. This architectural change, combined with half-precision support (FP16), reduces the running time of the optimizer step by up to $5\times$ compared to the standard Distributed Shampoo.

2. We investigate two new advanced linear algebraic approaches for computing matrix powers in the context of deep learning optimizers: the Newton-Denman-Beavers (Newton-DB) iteration (Higham, 2008) and Chebyshev polynomial approximation using Clenshaw's algorithm (Cody, 1970; Boyd, 2001). Motivated by the design of recent optimizers like Muon (Jordan et al., 2024), we explicitly aim to minimize the number of iterations required for these methods without degrading model performance. We show that our implementation of Newton-DB achieves lower validation perplexity than Coupled-Newton and standard Eigen-Value Decomposition when implemented into both Distributed Shampoo and our **DASH** block-preconditioned approach.

3. We provide a behavioral analysis of Newton-based iterations for matrix root computation. We demonstrate that the standard Frobenius norm scaling is suboptimal because it results in slower convergence, thereby requiring a higher number of iterations to reach the desired precision. Additionally, we offer intuitive explanations for the distinct convergence behavior observed in Coupled-Newton approaches compared to Newton-DB.

4. To address the Frobenius norm scaling limitations identified in our analysis, we introduce **multi-Power-Iteration**, an efficient half-precision implementation of the Power Iteration algorithm. This method robustly estimates the spectral radius (avoiding local maxima) to provide optimal scaling for the preconditioner blocks. This enables the Newton procedures to satisfy convergence criteria rapidly, further reducing the computational cost.

Our paper is structured as follows: Section 2 introduces notation, Shampoo optimizer and its features; Section 3 covers the inverse root methods, as well as the analysis of the iterative methods and multi-Power-Iteration; Section 4 describes the efficient blocking strategy; Section 5 describes experimental results and we finally conclude with related work and discussion in Section 6. We present the Chebyshev polynomial technique in Appendix A.

## 2. Preliminaries on Shampoo

This section introduces the notation used throughout the paper, a simplified version of the Shampoo optimizer and highlights the features (e.g. heuristics) used in Distributed Shampoo which our **DASH** implementation inherits, as well as new features we introduce in our work.

### 2.1. Notation

Throughout, we use $\theta_t = \{\theta_t^\ell \in \mathbb{R}^{m_\ell \times n_\ell}\}_{\ell=1}^{N_L}$ for the set of model parameters at optimization step $t$ with $N_L$ layers and $G_t = \{G_t^\ell \in \mathbb{R}^{m_\ell \times n_\ell}\}_{\ell=1}^{N_L}$ for the associated gradient; $f_{\theta_t}(x_t)$ for the model output that takes as input some sample data $x_t$ with associated label $y_t$; $\mathcal{L}(\hat{y}_t, y_t)$ for the loss function that requires as input the model prediction $\hat{y}_t$ and the target label $y_t$; $B$ for the block size used to split the gradient and subsequent states of Shampoo into blocks. In addition, we will use some abbreviations for the approaches we detail in our work: **EVD** for **E**igen-**V**alue **D**ecomposition, **CN** for **C**oupled-**N**ewton, **NDB** for **N**ewton-**D**enman-**B**eavers and **CBSHV** for **Ch**e**by**s**h**e**v** polynomials.

### 2.2. The Shampoo Optimizer

Given a gradient matrix $G_t \in \mathbb{R}^{m \times n}$, Shampoo computes left and right preconditioning matrices $L_t \in \mathbb{R}^{m \times m}$ and $R_t \in \mathbb{R}^{n \times n}$, incorporating products $G_t G_t^\top$ and $G_t^\top G_t$ respectively as an exponential moving average (EMA) parameterized by $\beta_{\text{LR}}$. Algorithm 1 presents pseudocode for Shampoo for a single matrix, where $\epsilon I$ is a regularization term for matrices $L_t$ and $R_t$. **DASH** inherits features from Distributed Shampoo, as described in Section 2.3.

### 2.3. Features Inherited from Distributed Shampoo

In this section we describe the techniques from Distributed Shampoo which we also integrated in our **DASH** implementation. We followed the pseudocode described in Shi et al. (2023, Algorithms 2 and 3). **Grafting.** Grafting (Agar-

---

**Algorithm 1** Simplified Outline of the Shampoo Optimizer

Initialize $L_0 = 0_{m \times m}, R_0 = 0_{n \times n}, \beta_{\text{LR}} \in (0, 1)$
**for** $t = 1$ **to** $T$ **do**
 $G_t = \nabla_\theta \mathcal{L}(f_{\theta_t}(x_t), y_t)$
 $L_t = \beta_{\text{LR}} \cdot L_{t-1} + (1 - \beta_{\text{LR}}) \cdot G_t G_t^\top$
 $R_t = \beta_{\text{LR}} \cdot R_{t-1} + (1 - \beta_{\text{LR}}) \cdot G_t^\top G_t$
 $\theta_{t+1} = \theta_t - \eta_t \cdot (L_t + \epsilon I_m)^{-1/4} \cdot G_t \cdot (R_t + \epsilon I_n)^{-1/4}$
**end for**

wal et al., 2020) is a technique introduced to transfer the learning rate schedule from another model. Specifically, it consists in using the unit-length direction from the current optimizer (Shampoo) and re-scale it to the norm of the other optimizer (Adam), for which we already have a tuned learning rate schedule. In short, this technique is called Adam grafting (applied to Shampoo). The preconditioners for both Shampoo and grafting method are updated based on the same sequence of iterates. Grafting is mandatory to have a numerically stable implementation for Shampoo. To briefly explain grafting, let $U_t = L_t^{-1/4} \cdot G_t \cdot R_t^{-1/4}$ be the Shampoo update and $P_t = G_t/(\epsilon + \sqrt{A_t})$ be the Adam-grafting direction (if EMA is enabled for $G_t$, one can use $M_t = \beta_G M_{t-1} + (1 - \beta_G)G_t$ instead). Then, the model is updated as $\theta_{t+1} = \theta_t - \eta_t \cdot s_t \cdot U_t$, where $s_t = ||P_t||_F/||U_t||_F$. For each layer, we implement blockwise grafting using Adam rule in our **DASH** as explained in Algorithm 2 in Distributed Shampoo.

**Load Balancing.** We implement the load-balancing approach explained in Algorithm 3 in Distributed Shampoo, which decides which GPU will process one specific layer. This is a greedy algorithm that sorts all layers by the total number of parameters in descending order and allocates each layer to the GPU that has the lowest load among all workers. The parameters are scattered on different workers to avoid redundant computations (according to the optimizer state partitioning strategy in ZeRO (Rajbhandari et al., 2020)). After all GPUs updated their own parameters assigned by the greedy procedure, we broadcast the updated parameters to synchronize the model across all workers.

### 2.4. New Features in DASH

**Lower-Precision Iterations.** Distributed Shampoo implemented the **CN** approach in float32 (FP32) precision. We introduce **CN** for FP16, which reduces the runtime of optimizer step by around $10\%$ compared to the FP32, with no degradation in validation perplexity. In the context of **CBSHV** (details in Appendix A), using FP16 *improves* the validation perplexity and reduces the running time, while for **NDB** it leads to numerical instabilities. We leave the investigation of FP16 for **NDB** for future work.

**Efficient Kernels.** Dion (Ahn et al., 2025) introduced ef-

ficient triton kernels to compute $X \cdot X^\top$, which we also employ in our optimizer to speed up computations for the **CN** approach.

**Memory Usage.** Our strategy to stack blocks in conjunction with the load balancing algorithm achieves better memory utilization across workers than Distributed Shampoo. For example, in our experiments for the 953M parameters in a setting with 8 GPUs, Distributed Shampoo uses 76GB memory per GPU, while our **DASH** uses 73 GB for higher rank workers and and 71 GB for lower rank workers.

## 3. Inverse Root Methods

This section first details the default numerical methods used in Shampoo, and then describes our additional Newton-DB approach for computing inverse roots $A^{-1/2}$ and $A^{-1/4}$ for a given matrix $A$, followed by a discussion on the importance of matrix scaling for the iteration convergence and our improvement to the Power-Iteration method. An additional technique for inverse roots based on Chebyshev polynomials can be found in Appendix A.

### 3.1. EVD: Eigen-Value Decomposition

Given a symmetric matrix $A \in \mathbb{R}^{n \times n}$ and $p \in \mathbb{R}$, a standard approach to compute matrix powers $A^p$ is to perform the **EVD** of $A$ as $A = Q\Lambda Q^\top$, where $Q$ are the eigenvectors of $A$ and $\Lambda$ are the eigenvalues of $A$, followed by $A^p = Q\Lambda^p Q^\top$. Despite its accuracy, **EVD** has the issue that its computation is hard to parallelize on GPUs, as the underlying algorithm is iterative, requiring building a Krylov subspace, re-orthogonalization of iterates and tridiagonalization (Lanczos, 1950; Ojalvo & Newman, 1970). The procedure has runtime $\Theta(n^3)$, becoming prohibitive for large matrices. Even though Distributed Shampoo (Shi et al., 2023) performs this operation in blocks of size $B$ as a trade-off heuristic to reduce complexity from $\Theta(n^3)$ to $\Theta(B^3)$ (for each of resulting blocks), **EVD** still incurs a large overhead for multiple such blocks because the inverse root is computed for each block sequentially. Therefore, iterative methods, such as Newton iterations based only on matrix-multiplications are fit for this scenario. While less accurate than **EVD**, they benefit from the high-performance primitives of optimized routines (matmul/gemm/bmm) powered by tensor-cores. In addition, our aim is to minimize the number of iterations for the Newton-like procedures to minimize the running time of **DASH**.

### 3.2. CN: The Coupled-Newton Iteration

The Coupled-Newton iteration (Higham, 2008, Equation 7.18) is implemented in Anil et al. (2020) as a faster alternative to the **EVD** approach. Given an input matrix $A \in \mathbb{R}^{n \times n}$ with eigenvalues $\lambda_i(A) \in [0, (p + 1)c^p]$ with the constant $c > 0$, it computes $A^{-1/p}$ iteratively, as described in Equations 1, 2 and 3. This requires defining two

sequences of matrices $X_k$ and $M_k$ with $X_k \xrightarrow{k \to \infty} A^{-1/p}$ and $M_k \xrightarrow{k \to \infty} I_n$.

$$X_0 = \frac{1}{c} I_n, \qquad M_0 = \frac{1}{c^p} A \tag{1}$$

$$C_k = \left(1 + \frac{1}{p}\right) I_n - \frac{1}{p} M_k \tag{2}$$

$$X_{k+1} = X_k C_k, \quad M_{k+1} = C_k^p M_k \tag{3}$$

The matrix $M_k$ is introduced for numerical stability and its closeness to the identity matrix $I_n$ within a desired tolerance can be used as a condition for early stopping. In Equation 2, the matrix $C_k$ is a linear combination of the current iterate $M_k$ and identity $I_n$ and it has to be raised to the $p$-th power to compute the next iterate $M_{k+1}$ in Equation 3.

The overhead per iteration for the **CN** approach is one matmul for the $X_k$ term, followed by one matmul for $p = 2$ and 2 matmuls for $p = 4$ to compute the $C_k^p$, plus one additional matmul to finally compute $M_{k+1}$. In total, the method requires 3 matmuls for $p = 2$ and 4 matmuls for $p = 4$.

### 3.3. NDB: The Newton-Denman-Beavers Iteration

The Newton-Denman-Beavers iteration (Higham, 2008, Equation 6.35) is an iterative procedure that computes both the square and inverse square roots of an input matrix $A \in \mathbb{R}^{n \times n}$, for which the condition $||I - A||_2 < 1$ holds. The **NDB** iteration is shown in Equations 4, 5 and 6, where it requires two sequences of matrices $Y_k$ and $Z_k$ with $Y_k \xrightarrow{k \to \infty} A^{1/2}$ and $Z_k \xrightarrow{k \to \infty} A^{-1/2}$. To compute inverse fourth root, we need two calls to the **NDB** procedure: from the first call we keep the square root, then input it to the second **NDB** call, keeping only the inverse square root which is actually the inverse fourth root (e.g. $(A^{1/2})^{-1/2} = A^{-1/4}$).

$$Y_0 = A, \qquad Z_0 = I_n \tag{4}$$

$$E_{k-1} = \frac{1}{2}(3I - Z_{k-1} Y_{k-1}) \tag{5}$$

$$Y_k = Y_{k-1} E_{k-1}, \quad Z_k = E_{k-1} Z_{k-1} \tag{6}$$

The overhead per iteration for the **NDB** approach is one matmul for each of the terms $E_k, Y_k, Z_k$, resulting in 3 matmuls per iteration. However, by inspecting the first iteration we observe two of these three matmuls are redundant. Specifically, under the initialization in Equation 4, the first iteration yields $E_1 = \frac{3}{2}I - \frac{1}{2}A$, $Y_1 = A \cdot E_1$ and $Z_1 = E_1$. The standard formulation computes $E_1$ and $Z_1$ via explicit matrix multiplications, despite their closed-form expression. To eliminate this redundancy, we directly initialize $E_1, Y_1$ and $Z_1$ to their values mentioned above and continue the iterative procedure from the second iteration onward, thus saving two matrix multiplications for the first iteration without altering the algorithmic result.

Since **NDB** computes powers $A^{\pm 1/2}$, chaining multiple calls can only compute powers $A^{\pm 1/2^k}$ with $k \in \mathbb{N}$. However, this is not a shortcoming in the context of Shampoo, since we only need $A^{-1/2}$ and $A^{-1/4}$.

### 3.4. Analyzing Matrix Scaling vs. Iteration Convergence

When using iterative procedures to compute inverse roots, the input matrix $A \in \mathbb{R}^{n \times n}$ requires some initial conditions to hold in order for the iteration to converge.

We will consider the initial condition for the **NDB** and **CN** methods as an example, i.e. $||I - A||_2 < 1$ for **NDB** and $||A||_2 < (p+1)c^p$ for **CN**, where $||X||_2$ is the operator norm of $X$ (i.e. the largest eigenvalue of the matrix $X$, denoted by $\lambda_{\max}(X)$). In order to have the same interval for the two methods and to simplify our discussion for the **CN** approach, we will use $(p + 1)c^p = 1$, i.e. $c = (1 + p)^{-1/p}$.

In theory, one should scale the matrix $A$ by $\lambda_{\max}(A)$ and the standard approach to compute $\lambda_{\max}(A)$ (besides inefficient **EVD**) is Power-Iteration. This iterative procedure computes an estimation of the eigenvector that corresponds to the largest eigenvalue (with a slight abuse of terms, we will call it the *largest eigenvector*), which is then plugged into the Rayleigh quotient, defined as $R_A(x) = (x^\top A x) / (x^\top x)$ to estimate the largest eigenvalue.

In Distributed Shampoo, the matrix is scaled by the Frobenius norm $||A||_F$, which is an upper bound on $\lambda_{\max}(A)$ and is computationally cheaper than Power-Iteration in their implementation. In practice, the gap between these quantities is quite large, with the Frobenius norm being larger than $\lambda_{\max}(A)$ by around $10 - 100\times$.

Our hypothesis is that these two choices of matrix scaling influence convergence, meaning that the iterative procedure will need more steps to converge to a target error, specifically when scaling the matrix by the Frobenius norm, which pushes eigenvalues towards zero. To validate this hypothesis, we designed the following numerical experiment: we input eigenvalues with different magnitudes in the interval $(0, 1)$ to **NDB** and **CN**, and record the number of iterations required to compute the inverse square root up to a fixed precision. Optionally, we also get the square root from the **NDB** approach.

Figure 1 confirms our hypothesis: smaller eigenvalues $x$ require more steps for **NDB** and **CN** to converge to the target values $x^{-1/2}$ and $x^{1/2}$ compared to the values closer to 1. The required number of steps to achieve a fixed precision is inversely proportional to the value of $x$. Since we use a fixed number of steps (e.g., 10) for both approaches in Shampoo, the error approximation for each eigenvalue depends on its magnitude: if the eigenvalue is small, using only 10 steps would yield a larger approximation error because in reality the iterative procedure requires more than 10 to

*Figure 1.* Number of steps required for **NDB** and **CN** to compute the square and inverse square roots of scalar numbers between 0 and 1 (in log-scale) up to precision $10^{-10}$ to emphasize the behavior for small eigenvalues.

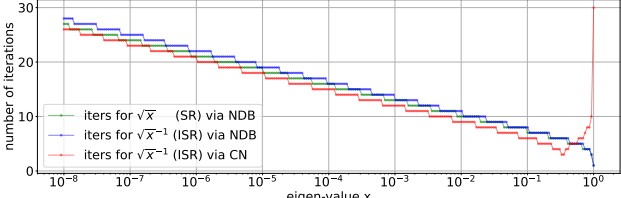

*Figure 2.* Number of steps required for **NDB** and **CN** to compute the square and inverse square roots of scalars between 0 and 1 (in linear scale) up to precision $10^{-10}$. We added a shift for **NDB** iterations to improve visibility on the y-axis.

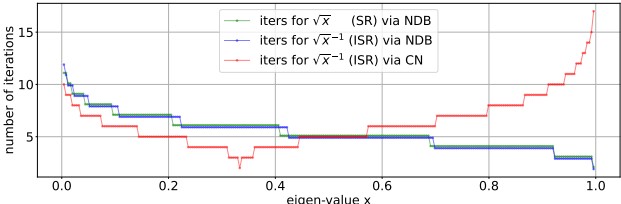

achieve the same error. Concretely, suppose the Frobenius norm is larger than the largest eigenvalue by $50\times$. Then, an eigenvalue $\lambda = $ 1e-2 that would normally require 5 steps to converge with **CN** would become $\lambda = $ 2e-4 after scaling it by frobenius norm, which would now require 15 steps with the same procedure.

Interestingly, the **CN** approach exhibits a significantly different behavior than **NDB** for the interval $x \in (0.3, 1)$. In Figure 2, we plot the x-axis on linear scale to emphasize the behavior of **NDB** and **CN** in this interval, where the values require more steps to converge for **CN**, with a peak of number of iterations around 1. This scenario will be encountered in practice when we use an accurate approximation of $\lambda_{\max}(A)$, a regime where **NDB** requires fewer steps than **CN**. This supports the usage of our **NDB** iteration in Shampoo instead of **CN**. In Section 5 we show that **NDB** consistently yields models with lower validation perplexity. A consequence of the above discussion is that a good approximation of $\lambda_{\max}(A)$ via Power-Iteration would have the effect of pushing the spectrum of $A$ towards 1 in a regime where we require fewer iterations. Recall that, for a real symmetric matrix $A \in \mathbb{R}^{n \times n}$ and any vector $x \in \mathbb{R}^n$, the Rayleigh quotient is $R_A(x) = (x^\top A x)/(x^\top x)$, which is upper-bounded by the largest eigenvalue $\lambda_{\max}(A)$, achieved when $x$ is the largest eigenvector. For any other vectors, Rayleigh quotient returns $\lambda(x) = R_A(x) < \lambda_{\max}(A)$, which is not enough to satisfy the convergence condition. Let $v_{\mathrm{PI}}$ be our estimation for the largest eigenvector $v^*$ obtained via Power-Iteration. Since $v_{\mathrm{PI}}$ is just an estimation of $v^*$, the associated largest eigenvalue via Rayleigh quotient is $\lambda_{\mathrm{PI}} = R_A(v_{\mathrm{PI}})$ that satisfies the condition $\lambda_{\mathrm{PI}} < \lambda_{\max}(A)$. Therefore, we chose to divide the matrix by $2\lambda_{\mathrm{PI}}$ instead of

Frobenius norm $||A||_{\mathrm{F}}$. This way, we make sure the estimation for the eigenvalue returned by Power-Iteration satisfies the convergence condition of **NDB** and **CN** approaches. In Section 5 we show that scaling by the Frobenius norm leads to numerical instabilities for **NDB** for larger block sizes, while scaling by Power-Iteration is stable, which supports our claim from this section.

### 3.5. Multi-Power-Iteration

In Distributed Shampoo, scaling the matrix by Frobenius norm is cheaper than the Power-Iteration. In contrast, our **DASH** implementation makes Power-Iteration computationally cheap specifically since we work with stacked blocks, which allows us to estimate all the largest eigenvectors at once.

It is known that the Power-Iteration can converge to an eigenvector that is not the largest (e.g. it does not correspond to the largest eigenvalue, but to a smaller one). To minimize the likelihood of this scenario, we can improve the estimation by using a *pool* of up to 16 or 32 starting vectors to estimate the largest eigenvectors in parallel. In the end, we choose the eigenvector with the largest Rayleigh quotient. We call this approach **multi-Power-Iteration**.

We emphasize that using multi-Power-Iteration instead of simple Power-Iteration does not increase the practical runtime. Matrix-vector multiplication is a memory-bound operation, with the memory transfers dominated by the size of $A$, so multiplying more vectors at the same time has negligible effect on the transfer size. By choosing the number of vectors to be a multiple of 16, we ensure the efficient use of tensor cores.

## 4. Blocking Strategy

In this section we present the blocking strategy implemented in Distributed Shampoo and how we turn it into our efficient strategy that powers up **DASH**. In turn, this leads to reduced running time for the optimizer step. In practice, **DASH** reduces the running time by up to $4 - 5\times$ compared to Distributed Shampoo.

Suppose we have a layer of shape $(m, n)$, where both $m$ and $n$ are perfectly divisible by $B$. Let $N_m = m/B$ and $N_n = n/B$ be the number of blocks for the $m-$ and $n-$dimension respectively and $N = N_m \cdot N_n$ be the total number of blocks of size $B \times B$ in the gradient matrix $G$. Therefore, we can write the gradient $G$ in a block-matrix form as:

$$
G = \begin{pmatrix} G_{11} & G_{12} & \cdots & G_{1N_n} \\ G_{21} & G_{22} & \cdots & G_{2N_n} \\ \vdots & \vdots & \ddots & \vdots \\ G_{N_m 1} & G_{N_m 2} & \cdots & G_{N_m N_n} \end{pmatrix}, \quad G_{ij} \in \mathbb{R}^{B \times B}
$$

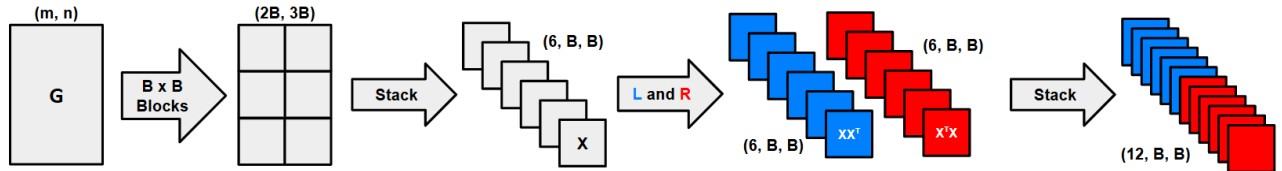

*Figure 3.* Stacking strategy in DASH.

### 4.1. Blocking Strategy in DASH

Our **DASH** implementation stacks all these blocks $G_{ij}$ into a 3D matrix $\text{block}(G) \in \mathbb{R}^{N \times B \times B}$ and applies batched operations for each inverse root procedure. This batching approach slightly improves the running time for **EVD** and it is especially fast for all other matrix-based approaches, such as the built-in **CN**, but also the **NDB** and **CBSHV** methods we introduce in this work. After creating $\text{block}(G)$, the Shampoo optimizer is implemented using the block matrix structure. Therefore, the products $GG^\top$ are computed using $\text{block}(G)$, where the transposition swaps the last two $B-$dimensions on the right.

In order to understand how this leads to practical speedups, let us consider a detailed example. First, we make the convention that $A \in \mathbb{R}^{N \times B \times B}$ can be written as $A \in (N, B, B)$ to improve the visibility of dimensions.

Let us consider block size $B = 1024$ and an embedding layer with vocabulary size $V = 32\,000$ and embedding size $E = 2048$, leading to a layer with shape $(V, E) = (32\,000, 2048)$, with number of blocks $N_m = \lfloor 31.25 \rfloor = 31$ and $N_n = 2$, with a total of $N = 62$ full blocks of shape $(B, B)$. Since $V \bmod B = 256$, we have two smaller blocks of shape $(256, 1024)$. In the end, the gradient $G$ is split into block matrices $G_{\text{full}} \in (62, 1024, 1024)$ and $G_{\text{rest}} \in (2, 256, 1024)$.

Next, we obtain the corresponding blocks for $L$ and $R$ as follows: $L_{\text{full}} \in (62, 1024, 1024)$, $L_{\text{rest}} \in (2, 256, 256)$, $R_{\text{full}} \in (62, 1024, 1024)$, $R_{\text{rest}} \in (2, 1024, 1024)$. Their corresponding inverse root matrices will have exactly the same shapes, which are the results of block-wise (or batch) matrix multiplications $GG^\top$ and $G^\top G$.

Our key observation is that matrices $L_{\text{full}}$, $R_{\text{full}}$ and $R_{\text{rest}}$ can be stacked together because they have the same shape $(B, B) = (1024, 1024)$. In contrast to Distributed Shampoo, which computes inverse root of each block matrix of shape $(B, B)$ sequentially, we can apply exactly the same inverse root procedure on the stacked matrices by performing only one call to our chosen inverse root procedure. **EVD** already supports batched matrices; the other approaches, such as **CN, NDB, CBSHV**, have to be modified to support batched matrix multiplications ($\text{bmm}$).

Concretely, we define the operator $\text{stack}(X, Y, Z)$ that stacks the matrices $X \in (N_X, B, B)$, $Y \in (N_Y, B, B)$ and $Z \in (N_Z, B, B)$ into $S \in (N_X + N_Y + N_Z, B, B)$. Therefore, we obtain the stacked matrix $S_{\text{full}} = \text{stack}(L_{\text{full}}, R_{\text{full}}, R_{\text{rest}}) \in (126, 1024, 1024)$ and $S_{\text{rest}} = L_{\text{rest}} \in (2, 256, 256)$. The classification head will have the resulting stacked matrices $S$ will have the $L_{\text{rest}}$ and $R_{\text{rest}}$ blocks swapped.

In Figure 3 we illustrate the stacking procedure for a case where the gradient has only 6 blocks of size $B$. The blocks are stacked into a 3D structure to efficiently compute the $L$ and $R$ matrices, which we can further stack together at the GPU level. This means the stacking procedure can be applied for all layers assigned to one GPU. This version is called DASHGPU in our implementation, which is the faster than DASHLAYERWISE that performs stacking per layer.

**Normalization Layers.** For a model with $N$ normalization layers, each of shape $E$, and a block size $B$, we stack all layers together into a tensor of shape $(N, E)$. Concretely, for $E = 2048$ and $B = 1024$, the gradient will have shape $(2N, B, 1)$ and the $L$ matrix and its inverse root will have shape $(2N, B, B)$. After computing the preconditioned gradient $U = L^{-1/2} \cdot G$ of shape $(2N, B, 1)$, we convert $U$ back to shape $(N, E)$ and then we update each of the $N$ normalization layers individually.

In our **DASH** implementation, we store the stacked 3-D tensors $S_{\text{full}}$, $S_{\text{rest}}$, $S_{\text{full}}^{-1/p}$ and $S_{\text{rest}}^{-1/p}$ for $p \in \{2, 4\}$. Instead, Distributed Shampoo stores two lists for $L$, $R$, $L^{-1/p}$ that contain individual blocks. It is important to mention that we need to store the inverse root buffers because Distributed Shampoo has the option to recompute them once at $f$ steps to alleviate the overhead of expensive procedures, such as **EVD**. In between two calls to the inverse root procedures, we have to use the stored buffers. In Section 5 we show that our implementation is fast even with $f = 1$. Running any inverse root procedure on a batch of matrices in our **DASH** implementation is faster than running on each individual matrix because we can benefit from the high throughput of the tensor-cores.

**Stacking benefits.** Stacking blocks of the $L, R$ matrices and for their corresponding inverse root avoids memory fragmentation, which occurs in Distributed Shampoo when individual blocks are stored in lists. Therefore, the matrix multiplications performed with stacked blocks (for non-

**EVD** approaches) become more efficient. Since working with stacked matrices is faster, this allows experimenting with lower-precision formats for matmuls (e.g. `float16`) and computing more accurate estimations for the largest eigenvalue used to scale the input matrix for the iterative procedures (more details in Section 3.4).

## 5. Experimental Results

We now show practical results for **DASH** compared to Distributed Shampoo (denoted by **DIST**) with respect to validation perplexity and running time of optimizer step.

**Setting.** We pretrain a Llama model with 953M parameters with embedding size $E = 2048$, sequence length 1024 and $2M$ token batch size with Chinchilla-optimal (Hoffmann et al., 2022) token counts (20 tokens/parameter) from the C4 dataset (Raffel et al., 2020), which results in 9089 optimization steps. We run 3 seeds for each experiment and show the average validation perplexity and running times. Since the converged runs are extremely stable when using Multi-PowerIteration, we omit standard deviations when reporting the results in this section and present the standard deviations in Appendix C. We present a preliminary set of results for Computer Vision in Appendix D.

**Learning Rate.** Since we use Adam grafting for both Shampoo implementations, we first run a grid search for AdamW and choose the learning rate that achieved the lowest validation perplexity to be used for our Shampoo runs. We found that the learning rate $\eta_* = $ 1e-3 performed the best across our grid $\eta \in \{$1e-4, 2e-4, 4e-4, 1e-3, 2e-3, 4e-3$\}$. The running time for the forward and backward passes will be the same for all models, and we only focus on measuring the running time of the optimizer step.

**Block Size.** We use preconditioner block sizes $B \in \{1024, 2048\}$ to validate the observation in Distributed Shampoo that increasing the block size leads to a closer approximation of full-matrix Shampoo. We posit that the block size $B$ should not be set to a larger value than the embedding size $E$, otherwise the blocks will be guaranteed to have rank at most $E$, which in turn adds noise to the spectrum of preconditioners. We experiment with preconditioner update frequencies $f \in \{1, 10\}$ for **EVD** and $f = 1$ for **CN** and **NDB**.

**Benchmarking.** Our results are summarized in Table 1, where we benchmarked the existing **DIST** implementation and **DASH**. Since our purpose is to speed up the computations in Shampoo, we compare the running time of optimizer step and validation perplexity in our evaluation. We are also interested in how inverse root methods **EVD**, **CN** and **NDB** influence these two metrics and how they behave in our **DASH** implementation.

**Overall Trends. DASH** matches **DIST** in almost all stable

configurations with respect to validation perplexity, while substantially reducing the running time of optimizer step by up to $4\times$ in one-to-one comparisons for each inverse root method and up to $5\times$ when comparing different inverse root methods. Interestingly, some setups achieved lower validation perplexity for **NDB** than **EVD**.

*Table 1.* Results for Llama-953M for Distributed Shampoo (**DIST**) and **DASH**. Abbreviations: **B** for block size, **IR** for inverse root method, **F** for preconditioner update frequency, **TIME** for optimizer step in milliseconds, **Scaling** for matrix normalization approach (**FRO** for Frobenius norm and and **PI** for Power-Iteration) and **Prec** for floating point precision **FP16/32** used for the iterations of the inverse root method. The **Time** column shows the running time of optimizer step for both Distributed Shampoo and **DASH** applied per layer as **LW** and per GPU (stacking the blocks from all layers allocated to one GPU) as **GPU**. The blue text shows the results obtained using our contribution.

| B | IR | F | Val PPL (↓) DIST | Val PPL (↓) DASH | Time (↓) DIST | Time (↓) LW | Time (↓) GPU | Scaling Prec |
|---|---|---|---|---|---|---|---|---|
| 2k | EVD | 1 | 11.72 | 11.73 | 2200 | **1747** | 1755 | - |
| | | 10 | 11.83 | 11.85 | 253 | 209 | 210 | - |
| | CN | 1 | 11.87 | **11.87** | 675 | 221 | **207** | FRO / 32 |
| | | 1 | 11.87 | **11.87** | **243** | 169 | **153** | FRO / 16 |
| | NDB | 1 | ✗ | **11.68** | ✗ | 279 | **264** | FRO / 32 |
| | | 1 | 11.73 | 11.72 | 355 | 284 | **267** | PIM / 32 |
| 1k | EVD | 1 | 11.80 | 11.81 | 3080 | 2850 | 2850 | - |
| | | 10 | 11.91 | 11.92 | 355 | 315 | 313 | - |
| | CN | 1 | 11.87 | 11.87 | 666 | 149 | **136** | FRO / 32 |
| | | 1 | 11.87 | 11.87 | **471** | 138 | **119** | FRO / 16 |
| | NDB | 1 | **11.76** | **11.77** | **558** | 188 | **174** | FRO / 32 |
| | | 1 | **11.69** | **11.68** | 740 | 194 | **177** | PIM / 32 |

**EVD.** For both block sizes, **DASH** achieves identical validation perplexity compared to **DIST**, with differences being $\approx 0.01$, while consistently reducing the runtime. Concretely, for $B = 2048$ and $f = 1$, **DASH** reduces running time from 2200ms to 1747ms for the LW-version ($1.26\times$ faster) and to 1755ms for the GPU-version ($1.25\times$ faster). For $f = 10$, the running time is reduced from 253ms to 210ms ($1.20\times$ faster) for both LW- and GPU-versions. Similar trend holds for $B = 1024$, where **DASH** improves runtime from 3080ms to 2850ms ($1.08\times$ faster) for $f = 1$ and from 355ms to 315ms ($1.13\times$) for $f = 10$ for both LW- and GPU-versions. We would like to emphasize that the running time for $f = 10$ is averaged across 10 consecutive optimizer steps using a running average with window size 10. The running time incurred for one preconditioning step is identical to the one for $f = 1$, but for the other 9 optimization steps in between two consecutive updates it takes roughly $\approx 35$ms to perform the step, which includes updating preconditioners and applying the inverse root that was cached after being previously computed. We observe that the GPU-version does not introduce any speedup compared

to the LW-version because the EVD itself is slow.

**CN.** For both FP32 and FP16 variants with Frobenius normalization, **DASH** (both LW and GPU variants) exactly matches the validation perplexity 11.87 of **DIST**, while yielding the largest relative speedups.

For $B = 2048$, **DASH** reduces the running time from 675ms to 221ms for the LW-version ($3.05\times$ faster) and to 207ms for the GPU-wise version ($3.26\times$ faster) for FP32. For FP16, the running time is reduced from 243ms to 169ms for LW-version ($1.44\times$ faster) and to 153ms for the GPU-version ($1.59\times$ faster). We can also cross-compare the running time of **DIST**-**CN**-FP32 of 675ms (previous best) with our contribution **DASH**-**CN**-FP16 of 153ms for the GPU-version ($4.41\times$ faster).

For $B = 1024$, **DASH** reduces the running time from 666ms to 149ms for LW-version ($4.47\times$ faster) and to 136ms for GPU-version ($4.90\times$ faster) for FP32. For FP16, the reduction in running time is from 471ms to 138ms for the LW-version ($3.41\times$ faster) and to 119ms for the GPU-version ($3.96\times$ faster). We can again cross-compare the running time of **DIST**-**CN**-FP32 of 666ms with **DASH**-**CN**-FP16 with the GPU-version of 119ms ($5.6\times$ faster), which is the highest relative improvement in running time in our evaluation. These results highlight that the block strategy and half-precision iterations provide a substantial reduction in runtime while preserving model performance.

In our experiments we omitted the Power-Iteration scaling for **CN** because it does not affect the results in any way. The high precision of FP16 format leads to no loss in validation perplexity and, as expected, it improves the running time. We also experimented with BF16 instead of FP16 and the **CN** method diverges. This is caused by the lower precision of the BF16 format, despite having the same range as FP32.

**NDB.** For both Frobenius norm and Power-Iteration normalizations, **NDB** consistently achieves lower validation perplexity than **CN**-FP32 (the built-in iterative inverse root method in **DIST**), while matching and even outperforming **EVD** across all preconditioner block sizes in both **DIST** and **DASH** implementations.

For $B = 2048$ and Power-Iteration normalization, **DASH** reduces the running time from 355ms to 284ms for the LW-version ($1.25\times$ faster) and to 267ms for the GPU-version ($1.33\times$ faster) and *achieves the same validation perplexity as **EVD** with $f = 1$*. However, the runs using Frobenius normalization failed for **DIST** across all seeds and therefore we skipped the results.

For $B = 1024$ and Frobenius normalization, **DASH** reduces the running time from 558ms to 188ms for the LW-version ($2.97\times$ faster) and to 174ms for the GPU-version ($3.21\times$ faster), while achieving lower validation perplexity

than **EVD**. However, when using Power-Iteration normalization, the runtime of **DIST** increases to 740ms because the Power-Iteration is not efficient when preconditioning the blocks sequentially. In contrast, **DASH** with Power-Iteration normalization requires 194ms for the LW-version ($3.81\times$ faster) and 177ms for the GPU-version ($4.18\times$ faster).

We would like to emphasize that scaling by Frobenius norm achieves 11.76 validation perplexity, while Power-Iteration achieves a significantly lower value of 11.68. This is a practical validation that Power-Iteration is a more accurate normalization choice than Frobenius, as shown in Section 3.4. In our experiments we omitted the FP16/BF16 results because **NDB** did not converge for these formats. We leave further investigations for future work.

**Optimal DASH Configuration.** Assume we wish to choose the best configuration of preconditioner block size $B$, inverse root method, precision and normalization to cross-compare the methods based on our results in Table 1. When optimizing for validation perplexity, one should definitely choose the **NDB** with Power-Iteration scaling approach we introduce, which is shown to achieve the lowest loss in our setting for the GPU-version. For practitioners who care about minimizing the optimizer runtime at all cost and afford trading some validation perplexity for a faster runtime, we suggest using **CN** with Frobenius normalization executed in FP16 using the GPU-version, which achieves the lowest runtime in our table. Regarding block size, we observed that increasing $B$ from 1024 to 2048 only implies increased running time without a significant improvement in validation perplexity. In fact, both block sizes achieve similar performances, but at different running times.

**Reduction in Overall Iteration Time.** In all our experiments we have the same runtime for the forward (FWP) and backward (BWP) passes and the only change is the inverse root method, which impacts the running time of optimizer step (OPT). We would like to express the reduction in OPT as a total reduction in time for the entire training run and we take the results for the preconditioner block size $B = 1024$ as an example. To keep the comparison simple, we will consider 9000 optimization steps instead of actual 9089. In our setup, each FWP requires $1000ms$, and each BWP requires 3000ms and on top of that we add the optimizer time. We chose the **DIST**-**EVD**-10 that requires 355ms, resulting in $10h\ 53m$ runtime and **CN**-FP16 that requires 138ms, resulting in $10h\ 21m$. Overall, this is a reduction of roughly $30m$ (5%) for training a 953M model.

## 6. Related Work and Discussion

Second-order methods can yield faster convergence, but can incur prohibitive quadratic runtime costs. The first approach to mitigate this was K-FAC (Martens & Grosse, 2015), which was a precursor to Shampoo (Gupta et al., 2018; Anil et al., 2020) and similar matrix adaptive opti-

mizers (Agarwal et al., 2019). The optimization community aimed at reducing the overhead of Shampoo in different ways. For example, prior work quantizes to 4-bit the eigenvectors obtained from Eigen-Value Decomposition (Wang et al., 2024) or even the preconditioning matrices themselves (Li et al., 2025) via Cholesky decomposition. Yen et al. (2023) focused on reducing the running time by performing a block low-rank decomposition using randomized SVD with shared basis in the context of AdaGrad by employing QR-decomposition to compute the dominant eigen-space, a procedure that has the same cubic complexity as the Eigen-Value Decomposition. On a similar note, SOAP (Vyas et al., 2024) computes Adafactor updates in the eigen-basis of Shampoo. Further, Xie et al. (2025) suggests using only the left preconditioner in Shampoo to reduce the overhead of inverse root methods, while Lin et al. (2025) recasts the Shampoo estimation as covariance estimation under KL-Divergence. Recently, Eschenhagen et al. (2025) performed an analytical deconstruction of Shampoo heuristics, suggesting that eigenvalue correction can replace grafting, which is compatible with our robust Power-Iteration approach.

**Discussion.** We presented DASH, a high-performance implementation of the Shampoo optimizer that bridges the gap between theoretical second-order guarantees and practical efficiency. By revisiting the algorithm from a systems perspective, we identified that the primary bottleneck was not mathematical complexity, but fragmented execution and numerical instability. Our contributions come in two distinct areas: numerical analysis (convergence, different solvers, multi-Power-Iteration) and system design (block execution to leverage TensorCores). Our approach opens several avenues for future work. For instance, the superior performance of Newton-DB suggests that dynamic solver selection might be beneficial: one could estimate the condition number of a block, and select the best/cheapest solver for it on the fly. A second interesting challenge is to stabilize our proposed Newton-DB iteration in the case of lower-precision execution. This could work by combining stochastic or error-correction methods, that have been shown to be effective for different preconditioners (Modoranu et al., 2023). A third direction is to validate our approach at even larger model and data scale, and leverage it in the context of tensor-parallel training.

## Impact Statement

This paper presents work whose goal is to advance the field of machine learning. There are many potential societal consequences of our work, none of which we feel must be specifically highlighted here.

## Acknowledgments

We would like to thank the Scientific Computing Department at ISTA for providing access to computational resources to develop this work and Denis Kuznedelev for proof-reading our manuscript. MS was supported by Research England under the Expanding Excellence in England (E3) funding stream, which was awarded to MARS: Mathematics for AI in Real-world Systems in the School of Mathematical Sciences at Lancaster University.

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

# Appendix

# Contents

# A. Chebyshev polynomials

In this section we explain how we use Chebyshev polynomials to approximate the inverse roots in Shampoo. Chebyshev polynomials is an infinite series of polynomials that represent an orthogonal basis to approximate functions. We will focus on Chebyshev polynomials of the first kind denoted by $T_n(x)$, which are defined recurrently as:

$$T_0(x) = 1 \tag{7}$$
$$T_1(x) = x \tag{8}$$
$$T_{n+1}(x) = 2 \cdot x \cdot T_n(x) - T_{n-1}(x) \tag{9}$$

To obtain the Chebyshev polynomials of the second kind, one should use $T_1(x) = 2x$. In our work we choose the first kind because it has better accuracy at the endpoints of the interval.

First, we will explain how this known result from linear algebra can be used to approximate a real function and then we will extend it to matrices.

## A.1. Chebyshev polynomials for real numbers

Given a function $f(x)$, we want to approximate it using Chebyshev polynomials. To do this, we need to decide how many terms we want to use. Let $d$ be the number of terms, which we call *the degree of the polynomial* because the largest power of the resulting polynomial will be at most $d$. Therefore, we want to approximate the function $f(x)$ as a linear combination of Chebyshev terms with coefficients $c_k \in \mathbb{R}$ as follows:

$$f(x) \approx \sum_{k=0}^{d} c_k \cdot T_k(x) \tag{10}$$

The Chebyshev polynomials can be evaluated using Clenshaw's algorithm (Clenshaw, 1955), explained step by step in Algorithm 3. One needs to fit the parameters $c_k$ only once, given the expression of $f(x)$, the degree $d$ and the number of points $N$, then perform at most $d + 1$ multiplications to evaluate the polynomial at each optimization step. Note the fitting procedure is cheap and it can be computed for large values of $N$, for example 1000 or 10 000 using only `numpy` (Harris et al., 2020).

According to (Cody, 1970), the Chebyshev series expansion for $f(x)$ is identical to the Fourier cosine series expansion for $f(\cos(\theta))$. Because of that, all inputs $x = \cos(\theta)$ belong to the interval $[-1, 1]$, thus our $x$ values have to be mapped to this interval.

**Important.** The Chebyshev polynomial ensures a good approximation only for the numbers in the interval $[a, b]$. Given $x \in [a, b]$, we can compute $x^{-1/p}$ by a linear mapping of $x$ to the interval $[-1, 1]$, followed by calling the Clenshaw's algorithm. Since the coefficients were fitted for the interval $[a, b]$, they implicitly embed the mapping.

---

**Algorithm 2** Fitting Coefficients for Chebyshev Polynomial

---

**Input:** function $f(x)$, degree $d$, number of points $N$
**Output:** Chebyshev coefficients $c \in \mathbb{R}^{d+1}$
$v \leftarrow [0, 1, \cdots, N-2, N-1]$ // points used to fit coefficients
$\theta \leftarrow (2v + 1)\frac{\pi}{2N}$ // compute cosine frequencies
$t \leftarrow \cos(\theta)$
$x \leftarrow \frac{1}{2}(b-a)t + \frac{1}{2}(b+a)$ // convert $t_i \in [-1, 1]$ to $x_i \in [a, b]$
$c = 0_{d+1}$ // initialize $d + 1$ entries with zero, that will store the $d$ coefficients $c_k$ from 0 to $d$ inclusively
$f_x = f(x)$ // compute $f(x)$ only once
**for** $k = 0$ **to** $d$ (inclusively) **do**
    $c_k = \frac{2}{N} f_x^\top \cos(k \cdot \theta)$ // scalar product between $f(x) = x^{-1/p}$ and the vector $\cos(k \cdot \theta)$
**end for**
$c_0 \leftarrow \frac{1}{2}c_0$ // make the cosine transform orthogonal
**return** $c \in \mathbb{R}^{d+1}$

---

---

**Algorithm 3** Clenshaw's Algorithm to Evaluate the Chebyshev Polynomial (scalar case)

---

**Input:** input value $x \in [a, b]$, Chebyshev coefficients $c \in \mathbb{R}^{d+1}$ fitted for $f(x)$ and $[a, b]$
**Output:** estimation $x^{-1/p}$ via Chebyshev polynomial
$b_{d+2} \leftarrow 0 \in \mathbb{R}$
$b_{d+1} \leftarrow 0 \in \mathbb{R}$
**for** $k = d$ down to 0 (inclusively) **do**
$\quad b_k \leftarrow 2 \cdot x \cdot b_{k+1} - b_{k+2} + c_k$
**end for**
$\hat{x} \leftarrow b_0 - x \cdot b_1$
**return** $\hat{x} \approx f(x) = x^{-1/p}$

---

## A.2. Chebyshev polynomials for matrices

Chebyshev polynomials can be extended from the real case to the space of matrices by doing a few changes. Let $A \in \mathbb{R}^{n \times n}$ and we want to compute $f(A) = A^{-1/p}$, which we can do in a few steps.

Once we decide the degree of the polynomial, we need to specify the interval $[a, b]$ to fit the coefficients. In the matrix case, we care about the eigenvalues of matrix $A$ and we need to make sure they lie in the interval $[a, b]$. To be in line with the Eigen-Value Decomposition approach, we choose $[a, b] = [\epsilon, 1 + \epsilon]$ and we fit the Chebyshev coefficients for this interval.

Next, given the input matrix $A$ with eigenvalues in $[\lambda_{min}(A), \lambda_{max}(A)]$, we need to obtain a matrix $S$ with eigenvalues in the interval $[-1, 1]$ before calling Clenshaw's algorithm on $S$ to estimate $A^{-1/p}$. We scale the matrix $A$ by Frobenius norm (or Power-Iteration) to have the eigenvalues in the interval $[0, 1]$ and then we map it to $[-1, 1]$ by multiplying by 2 and subtracting the identity matrix, as presented in Algorithm 4, where we choose to scale the matrix $A$ via Frobenius norm:

---

**Algorithm 4** Clenshaw's Algorithm to Evaluate the Chebyshev Polynomial (matrix case)

---

**Inputs:**
- input matrix $A \in \mathbb{R}^{n \times n}$ with eigenvalues in $[\lambda_{min}(A), \lambda_{max}(A)]$
- coefficients $c \in \mathbb{R}^{d+1}$ fitted for $f(x)$ and $[\epsilon, 1 + \epsilon]$
**Output:** estimation of $A^{-1/p}$ via Chebyshev polynomial
$S = 2\frac{A}{||A||_F} - I_n \text{ // } \lambda_i(S) \in [-1, 1]$
$B_{d+2} \leftarrow 0_n \in \mathbb{R}^{n \times n}$
$B_{d+1} \leftarrow 0_n \in \mathbb{R}^{n \times n}$
**for** $k = d$ down to 0 (inclusively) **do**
$\quad B_k \leftarrow 2 \cdot S \cdot B_{k+1} - B_{k+2} + c_k \cdot I_n$
**end for**
$\hat{A} \leftarrow B_0 - S \cdot B_1$
**return** $\hat{A} \approx f(A) = A^{-1/p}$

---

Note this algorithm requires $d + 2$ matrix multiplications. By carefully inspecting the iteration, we can perform some optimizations.

The first optimization consists in observing that for steps $k \in \{d, d - 1\}$ there are some redundant multiplications with zero matrices. Instead of spending FLOPs on multiplications with zero, one can manually compute the terms $B_d$ and $B_{d-1}$ and reduce the number of iterations (thus matmuls) by 2 in the for-loop. Concretely, for steps $d$ and $d - 1$, we obtain $B_d = c_d \cdot I_n$ and $B_{d-1} = 2 \cdot c_d \cdot S + c_{d-1} \cdot I_n$. Therefore, we can now initialize $B_d$ and $B_{d-1}$ with these quantities and then iterate $k \in \{d - 2, d - 1, ..., 1, 0\}$.

The second optimization consists in observing that the last step $\hat{A} = B_0 - S \cdot B_1$ is also redundant. By plugging in the recurrence formula $B_0 = 2 \cdot S \cdot B_1 - B_1 + c_0 \cdot I_n$ into the expression of $\hat{A}$, we obtain:

$$\hat{A} = B_0 - S \cdot B_1 \tag{11}$$
$$= (2 \cdot S \cdot B_1 - B_2 + c_0 \cdot I_n) - S \cdot B_1 \tag{12}$$
$$= S \cdot B_1 - B_2 + c_0 \cdot I_n \tag{13}$$

In the final expression we observe that we do not explicitly need to compute all terms including $B_0$, but we can stop at $B_1$. Therefore, we can now iterate $k \in \{d-2, d-1, ..., 2, 1\}$. In the end, we perform $d-2$ matrix multiplications in the for-loop and another one after the for-loop to compute the final approximation, resulting in a total of $d-1$ matrix multiplications for a Chebyshev polynomial with degree-$d$. The advantage here is that we can achieve the complexity of $d$ matmuls and actually use a polynomial of degree $d+1$. Conversely, if we choose degree $d$, we only pay for $d-1$ matmuls.

We present the final version in Algorithm 5.

---

**Algorithm 5** Optimized Clenshaw's Algorithm to Evaluate the Chebyshev Polynomial (matrix case)

---

**Inputs:**
- input matrix $A \in \mathbb{R}^{n \times n}$ with eigenvalues in $[\lambda_{min}(A), \lambda_{max}(A)]$
- coefficients $c \in \mathbb{R}^{d+1}$ fitted for $f(x)$ and $[\epsilon, 1 + \epsilon]$
**Output:** estimation of $A^{-1/p}$ via Chebyshev polynomial
$S = 2 \frac{A}{||A||_F} - I_n$
$B_d \leftarrow c_d \cdot I_n \in \mathbb{R}^{n \times n}$
$B_{d-1} = 2 \cdot c_d \cdot S + c_{d-1} \cdot I_n \in \mathbb{R}^{n \times n}$
**for** $k = d-2$ down to 1 (inclusively) **do**
    $B_k \leftarrow 2 \cdot S \cdot B_{k+1} - B_{k+2} + c_k \cdot I_n$
**end for**
$\hat{A} \leftarrow S \cdot B_1 - B_2 + c_0 \cdot I_n$
**return** $\hat{A} \approx f(A) = A^{-1/p}$

---

### A.3. Numerical Precision.

Clenshaw's algorithm used to evaluate the Chebyshev polynomial requires high precision for the iterates $B_k$, meaning that `float32` is required for storage. However, one can benefit from a two times higher throughput from tensor-cores on Nvidia GPUs by performing the multiplications in half precision and do accumulation in `float32`. We found that applying this technique with `float16` improves the results compared to `float32`.

Concretely, we store $B_k$ in `float32` and $S$ in `float16` and at each iteration $k \in \{d-2, d-1, ..., 2, 1\}$ to compute $B_k$, we convert $B_{k+1}$ to `float16` and perform the matmul $S \cdot B_{k+1}$ in `float16`, with the important mention that the accumulation should be done in `float32`. This is a very important part of this optimization, otherwise the matrix multiplication procedure would return a `float32` buffer. There are a few more optimizations one can do here, such as avoiding repeated allocations in the `bmm` call and we let this for future work.

### A.4. Experiments.

Below we provide a limited set of results for the **CBSHV** polynomial that serve as a preliminary evaluation for this method.

We test our **CBSHV** method with degree $d = 60$ for the Llama-373M model with embedding size $E = 1024$ and block size $B = 1024$ when trained with a batch size of 2 million tokens in both Distributed Shampoo (**DIST**) and our **DASH**, where we update the normalization layers using Adam (**DASH**-A).

**DIST** runs performed in `float32` precision lead to a much higher validation perplexity of 18.6, in contrast to only 16.15 for the `float16` version for both Frobenius and Power-Iteration normalizations. Unfortunately we do not have a rigorous explanation for why this happens, as the performed computations are similar for both Distributed Shampoo and **DASH**. It is important to mention that the running time of the `float16` implementation is higher than for `float32`, which is against our hypothesis that we can benefit from the higher throughput of tensor-cores. The explanation for this matter is the sequential computation of inverse roots performed in DIST, which is inefficient even at this small model scale and it serves as the motivation of our work to reduce the running time.

On the other side, **DASH**-A has a more consistent validation perplexity. For `float32`, both Frobenius norm and Power-Iteration normalization converge, with Power-Iteration achieving higher validation perplexity. However, the running time is much lower, under 90 milliseconds per optimizer step, which is an improvement of a bit less than $2\times$. For `float16`, scaling by Frobenius norm converges with a running time of 76 milliseconds. This is an indication that our **DASH** implementation allows lower-precision improvements. On the other side, using Power-Iteration in this context did not converge for the three

seeds we used.

Since both **DIST** and **DASH**-A have some drawbacks at this small scale, we decided not to add it to the main body of our work. However, we believe it is important to bring this technique to the community's attention as it can pave the way for improvements and potentially other applications than Shampoo optimizer.

Next, we are going to show some preliminary results for the larger model.

*Table 2.* Results for **CBSHV** with degree $d = 60$ for Llama-373M, where Normalization Layers are updated using Adam (**DASH**-A). We update the preconditioners at each optimization step ($f = 1$). Time is in milliseconds.

| IMPL | VAL PPL | TIME | INFO |
|------|---------|------|------|
| DIST | 18.60 | 168 | FRO / FP32 / $d = 60$ |
|      | 18.63 | 176 | PI / FP32 / $d = 60$ |
|      | 16.15 | 201 | FRO / FP16 / $d = 60$ |
|      | 16.15 | 193 | PI / FP16 / $d = 60$ |
| DASH-A | 16.14 | 80 | FRO / FP32 / $d = 60$ |
|      | 16.22 | 84 | PI / FP32 / $d = 60$ |
|      | 16.15 | 76 | FRO / FP16 / $d = 60$ |
|      | ✗ | ✗ | PI / FP16 / $d = 60$ |

*Table 3.* Results for **CBSHV** with degree $d \in \{40, 60, 100\}$ for Llama-953M, where Normalization Layers are updated using Adam. We update the preconditioners at each optimization step ($f = 1$). Time is in milliseconds.

| IMPL | BLOCK | VAL PPL | TIME | INFO |
|------|-------|---------|------|------|
| DIST | 2048 | 12.00 | 580 | FRO / FP32 / $d = 100$ |
|      |      | 12.04 | 538 | FRO / FP16 / $d = 100$ |
|      | 1024 | 11.88 | 1056 | FRO / FP32 / $d = 100$ |
|      |      | 11.87 | 957 | FRO / FP16 / $d = 100$ |
| DASH-A | 2048 | 12.10 | 330 | FRO / FP32 / $d = 60$ |
|      |      | 12.10 | 262 | FRO / FP16 / $d = 60$ |
|      | 1024 | 11.99 | 242 | FRO / FP32 / $d = 60$ |
|      |      | 11.98 | 228 | FRO / FP16 / $d = 60$ |
|      |      | 12.11 | 162 | FRO / FP16 / $d = 40$ |

# B. Additional Improvements to Distributed Shampoo

In this section we provide a few observations for the Distributed Shampoo implementation that arised during the preparation of our work.

## B.1. Regularization (dampening) for EVD

In the context of using **EVD** for inverse roots, the preconditioner blocks have to be regularized in order for the Eigen-Value Decomposition procedure to converge. This is mandatory because **absolutely all** blocks have to converge, otherwise the entire training will fail. As explained in Section 3.2.1 from the original Distributed Implementation (Shi et al., 2023), the authors explain how they apply regularization to each preconditioner block, which we reproduce here:

---

**Symmetric Eigendecomposition Approach for Computing Root Inverse**
Given $L \in \mathbb{R}^{n \times n}$ (or $R$), perturbation $\epsilon > 0$, and desired exponent $r$.

1. Compute symmetric eigendecomposition $\lambda, Q \leftarrow \texttt{eigh}(L)$ where $\lambda \in \mathbb{R}^n$ and $Q \in \mathbb{R}^{n \times n}$.

2. Compute $\lambda_{\min} \leftarrow \min_i \lambda_i$.

3. Compute $\lambda_{new} \leftarrow \lambda - \min(\lambda_{\min}, 0)1 + \epsilon 1$.

---

> 4. Form and return matrix root inverse $L_{inv} \leftarrow Q \operatorname{diag}(\lambda_{new}^{-r}) Q^T$.

In the Distributed Shampoo implementation, the authors implement step 1 as $\lambda, Q \leftarrow \texttt{eigh}(L + \epsilon I_n)$ and then proceed with steps 2, 3 and 4. Our observation is that the eigenvalues $\lambda$ already contain the regularization $\epsilon$ and we state it should be subtracted from $\lambda$ after the first step, otherwise in step 3 it will be added again, which would result in an inconsistent regularization because some entries will be incrased by $2\epsilon$ and others by less than $\epsilon$.

Let's consider a numerical example for regularization value $\epsilon = 10^{-10}$. Suppose the matrix $L$ has the smallest eigenvalue $\lambda_{min}(L) = 10^{-12}$ **before** adding regularization in the `eigh` call. Step 1 will call $\lambda, Q \leftarrow \texttt{eigh}(L + 10^{-10}I_n)$ and therefore in step 2 we will get $\lambda_{min} = 10^{-12} + 10^{-10}$. In step 3, the value of $\lambda_{min}$ will be set to $\lambda_{min}^{new} = \lambda_{min} - \min(\lambda_{min}, 0) + \epsilon = \lambda_{min} + \epsilon = 10^{-12} + \mathbf{2} \cdot 10^{-10}$, which is equivalent to adding $2\epsilon$. When the matrix $L$ is low-rank, it is likely that $\lambda_{min} < 0$. In this case, $\min(\lambda_{min}, 0) = \lambda_{min} < 0$ and therefore $\lambda_{min}^{new} = \epsilon$. The other eigenvalues will be increased by $\epsilon - \lambda_{min}$.

In summary, the Distributed Shampoo implementation does not increase all eigenvalues by $\epsilon$, but by $2\epsilon$ for $\lambda_{min}$ and by $\epsilon - \lambda_{min}$ for the other eigenvalues $\lambda_i \neq \lambda_{min}$.

### B.2. Our Dampening Heuristics.

To avoid the inconsistencies in applying the regularization in Distributed Shampoo, we propose three new heuristics. By default, we perform **EVD** for $L$ as $\lambda, Q \leftarrow \texttt{eigh}(L + \epsilon I_n)$ and then subtract $\epsilon$ from $\lambda$ as $\lambda \leftarrow \lambda - \epsilon$, which we call *corrected spectrum*. Since the small magnitude eigenvalues will be amplified when we raise them to power $-1/p$, we believe it is important to filter them out.

**Shifted-ReLU heuristic.** We apply ReLU to the corrected spectrum to zerorize the entries that are smaller than $\epsilon$, which is equivalent to doing $\lambda \leftarrow \text{RELU}(\lambda - \epsilon)$. This way, we filter out the noise in the corrected spectrum and therefore keep only the entries we consider significant (e.g. larger than $\epsilon$). This is essentially the same as performing an implicit low-rank selection of eigenvectors in $Q$ that is $\epsilon$-dependent. Therefore, when computing $\lambda^{-1/p}$, we can raise to power $-1/p$ only the entries in the final ReLU-shifted $\lambda$ that are larger than zero and the multiplication $Q(\lambda^{-1/p})Q^\top$ will actually obtain a rank-$r_\epsilon$ inverse, where $r_\epsilon = \sum_{i=1}^{n} I(\lambda_i > 0)$ is the number of non-zero eigenvalues after applying shifted-ReLU.

**ABS-based heuristic.** Instead of applying ReLU on the corrected spectrum to remove the negative eigenvalues, we can also apply absolute value function to make sure all eigenvalues are positive. Optionally, we can also add $\epsilon$ to make sure all eigenvalues have a lower bound.

We test the three heuristics (Shifted-RELU, ABS-ADD, SHMP-the original used in DS) on Llama-373M using frequency 1 and on Llama-953M with frequency 10 on DASH with EVD.

| OPTIMIZER | METHOD | MODEL | BLOCK | FREQ | HEURISTIC | VAL PPL | RUNTIME |
|-----------|--------|-------|-------|------|-----------|---------|---------|
| | | | | | SHMP | 15.893 | |
| DASH | EVD | 373M | 1024 | 1 | ABS-ADD | 15.886 | 4H |
| | | | | | RELU | 15.876 | |
| | | | | | SHMP | 11.850 | |
| DASH | EVD | 953M | 2048 | 10 | ABS-ADD | 11.856 | 10H 50M |
| | | | | | RELU | 11.866 | |

*Table 4.* Comparison of heuristics on DASH with EVD.

The results show there is no heuristic that clearly stands out, as the validation perplexities are similar. We believe a better setting would be to run Llama-953M with frequency 1. Unfortunately, our compute was limited and we could not afford a 17 hours run in this setting.

## C. Standard Deviations for DASH

During the preparation of our work we ran 3 seeds only for a few experiments we reported in Table 1 because the runs are quite stable, especially when using Multi-PowerIteration. We report the exact results we obtained for each seed below:

| B | OPTIMIZER | METHOD | FREQ | NORM | SEED | VAL PPL | AVG PPL | STD PPL |
|---|---|---|---|---|---|---|---|---|
| | | | | | 42 | 11.780 | | |
| | DIST | NDB | 1 | FRO | 666 | 11.743 | 11.764 | .019 |
| | | | | | 2408 | 11.769 | | |
| 1K | | | | FRO | 666 | 11.700 | 11.692 | .011 |
| | | | | | 2408 | 11.684 | | |
| | DASH | NDB | 1 | | 42 | 11.726 | | |
| | | | | PIM | 666 | 11.726 | 11.726 | .0005 |
| | | | | | 2408 | 11.725 | | |
| | | | | | 42 | 11.772 | | |
| 2K | DASH | NDB | 1 | PIM | 666 | 11.773 | 11.771 | .002 |
| | | | | | 2408 | 11.768 | | |

# D. DASH for Computer Vision

Our DASH optimizer is mainly designed for Transformer models, which contains only 1D and 2D layers and we haven't tested it on other architectures than Transformers.

The main reason for this is because NewtonDB (NDB) procedure can compute inverse roots that are powers of two, such as $\pm 1/(2^k)$. For 3D or 4D layers, we would need to apply Algorithm 2 in the original Shampoo paper (**?**), which computes powers $-1/(2 \cdot k)$ for the $k$-th dimension. For $k \in \{1, 2\}$, DASH rule is identical to DS and for $k \in \{3, 4\}$, DS computes power $-1/6$ for the 3rd dimension and $-1/8$ for the 4th dimension.

We test DASH on two types Vision Transformers (Tiny-5M, Small-21M) on two subsets of ImageNet called ImageNette (10 easy classes) and ImageWoof (10 breeds of dogs - difficult).

In ViTs, the embedding layer is 3D with shape $(E, 3, 16, 16)$, where $E$ is embedding size. DASH converts the 3D embedding layer to 2D of shape $(E, 768)$ by merging the remaining dimensions ($768 = 3 * 16 * 16$).

When we integrate NDB into DS, we compute power $-1/2$ for the first dimension and power $-1/4$ for all other dimensions.

In the table below we provide the results for these experiments. Briefly, NDB is more accurate than CN when plugged to DASH and DS. The accuracy is especially higher for DS-NDB on ImageWoof/ViT-S. Despite being more accurate than DASH, DS is 10x slower. These results show the inverse root used in the preconditioner does not impact the results and NDB method stands out even in this setting where 3D and 4D tensors are converted to 2D in the context of DASH.

| DATASET | MODEL | OPTIMIZER | METHOD | TEST ACC(%) | OPT STEP (MS) |
|---|---|---|---|---|---|
| | | DASH | CN | 81.62±.65 | 9 |
| | | | NDB | 82.45±.42 | 9 |
| | ViT-T | DIST | CN | 81.05±1.10 | 104 |
| NETTE | | | NDB | 84.06±.55 | 140 |
| | | DASH | CN | 81.44±.38 | 15 |
| | | | NDB | 83.14±.56 | 15 |
| | ViT-S | DIST | CN | 81.43±.38 | 116 |
| | | | NDB | 84.12±.49 | 145 |
| | | DASH | CN | 66.86±.19 | 9 |
| | | | NDB | 69.86±.28 | 9 |
| | ViT-T | DIST | CN | 65.95±.85 | 105 |
| WOOF | | | NDB | 72.90±.11 | 140 |
| | | DASH | CN | 64.14±1.02 | 15 |
| | | | NDB | 70.28±.12 | 15 |
| | ViT-S | DIST | CN | 63.80±.72 | 115 |
| | | | NDB | 72.08±.44 | 145 |

*Table 5.* Results on Vision Transformers using DASH and DistributedShampoo (DIST) with CN and NDB inverse root methods.

