# OpenReview forum: "DASH: Faster Shampoo via Batched Block Preconditioning and Efficient Inverse-Root Solvers"
_ICML.cc/2026/Conference — ICML 2026 regular_

### Official Review · Reviewer_H8GD · 2026-03-06

**Soundness:** 2
**Presentation:** 2
**Significance:** 2
**Originality:** 2
**Overall Recommendation:** 4
**Confidence:** 3

**Summary:**

This paper introduces DASH (Distributed Accelerated SHampoo), a high-performance implementation of the Shampoo optimizer designed to significantly reduce its computational overhead while preserving numerical precision and model performance. The paper evaluates DASH on Llama models (373M and 953M parameters) trained on C4, showing that DASH matches or improves validation perplexity while substantially reducing optimizer runtime (up to 4.83× speedup) and memory usage compared to Distributed Shampoo.

The main contributions are the following:

1- A novel GPU-efficient approach that replaces sequential loops over preconditioner blocks with batched operations on 3-D tensors, enabling parallel processing and better GPU utilization. This, combined with half-precision support (FP16), reduces optimizer step time by up to 5× compared to Distributed Shampoo.

2- Investigation and implementation of two advanced linear algebraic methods—Newton-Denman-Beavers (NDB) iteration and Chebyshev polynomial approximation—for computing matrix inverse roots. The authors show that NDB achieves lower validation perplexity than both Coupled-Newton (CN) and eigenvalue decomposition (EVD).

3- A robust half-precision implementation of power iteration that accurately estimates spectral radii for optimal matrix scaling, addressing the limitations of Frobenius norm scaling, which leads to slower convergence of iterative methods.

4- Theoretical and empirical analysis of Newton-based iterations, demonstrating why Frobenius norm scaling is suboptimal and explaining convergence differences between CN and NDB.

5- Integration of efficient kernels, load balancing, and memory optimizations that reduce memory fragmentation and improve overall throughput.

**Compliance With Llm Reviewing Policy:**

Affirmed.

**Final Justification:**

The authors addressed my concerns, so I increased the score.

**Key Questions For Authors:**

1- The authors propose new dampening heuristics for EVD (Shifted-ReLU, ABS-based). How do these affect convergence and final perplexity compared to the original Distributed Shampoo approach? Did you observe cases where one heuristic consistently outperforms others?

2- The results are for 953M parameters. How does DASH scale to multi-billion parameter models with tensor parallelism? Does the stacking strategy work efficiently when preconditioner blocks are distributed across devices, or are there communication overheads that become bottlenecks?

**Limitations:**

1- The method is evaluated only on transformer-based language models; generalization to other architectures (CNNs, etc.) is not shown.

2- While the paper notes that "converged runs are extremely stable" and omits standard deviations, reporting variance would strengthen claims of reliability.

3- Experiments use 373M and 953M parameters. Performance on smaller models (where overhead might dominate) or larger models (where communication might become a bottleneck) is not shown.

**Strengths And Weaknesses:**

Strengths

1- The experimental methodology is rigorous: controlled comparisons between Distributed Shampoo and DASH under identical conditions.

2- Second-order optimizers like Shampoo offer theoretical advantages but are often impractical due to computational overhead. DASH directly addresses this gap, making Shampoo viable for large-scale training.

3- DASH modifies only the optimizer implementation, not the model architecture or training pipeline, making adoption easier.

Weaknesses

1- While the paper observes that block size 1024 performs similarly to 2048 at lower cost, there is no theoretical analysis of how block size affects preconditioner quality or convergence rate.

2- All experiments use C4. Results on other domains (image, graph, speech) or datasets would strengthen generalizability claims.

3- Unlike some optimizer papers that provide regret analysis or convergence guarantees, DASH focuses on empirical speedups without theoretical guarantees about optimization performance.

---

> ### Author Rebuttal · Authors · 2026-03-31
>
> We would like to thank the reviewer for their feedback. We address your concerns below.
>
> **Weakness 1**
>
> Providing a theoretical analysis for how the block size affects the preconditioner quality is not the focus of our work. We inherit this default heuristic proposed in the well established Distributed Shampoo (DS) optimizer and focus on improving the runtime of optimizer step and validation perplexity.
>
> **Weakness 2**
>
> Please check our answer for Limitation 2 below.
>
> **Weakness 3**
>
> Please check our answer for **Question 2** of **Reviewer 9Xeh**.
>
> **Question 1**
>
> We test the three heuristics (Shifted-RELU, ABS-ADD, SHMP-the original used in DS) on Llama-373M using frequency 1 and on Llama-953M with frequency 10 on DASH with EVD.
>
> |opt|method|model|block|freq|heuristic|val ppl|runtime|
> |-|-|-|-|-|-|-|-|
> |DASH|EVD|373M|1024|1|SHMP|15.893|4h|
> ||||||ABS-ADD|15.886||
> ||||||RELU|15.876||
> ||
> |||953M|2048|10|SHMP|11.850|10h50m|
> ||||||ABS-ADD|11.856||
> ||||||RELU|11.866||
>
> The results show there is no heuristic that clearly stands out, as the validation perplexities are similar. We believe a better setting would be to run Llama-953M with frequency 1. Unfortunately, our compute was limited and we did afford a 17 hours run.
>
> **Question 2**
>
> Our experiments in the paper are limited for the DDP scenario, however we are currently working on an FSDP-compatible version. DASH can independently update blocks extracted from the local shard, just like AdamW, without performing any communication at the optimizer level.
>
> **Limitation 1**
>
> Our DASH optimizer is mainly designed for Transformer models, which contains only 1D and 2D layers and we haven’t tested it on other architectures than Transformers.
>
> The main reason for this is because NewtonDB (NDB) procedure can compute inverse roots that are powers of two, such as $\pm 1/(2^k)$. For 3D or 4D layers, we would need to apply Algorithm 2 in the original Shampoo paper [1], which computes powers $-1/(2 \cdot k)$ for the k-th dimension. For $k \in \\{1, 2\\}$, DASH rule is identical to DS and for $k \in \\{3, 4\\}$, DS computes power $-1/6$ for the 3rd dimension and $-1/8$ for the 4th dimension.
>
> We test DASH on two types Vision Transformers (Tiny-5M, Small-21M) on two subsets of ImageNet called ImageNette (10 easy classes) and ImageWoof (10 breeds of dogs - difficult).
>
> In ViTs, the embedding layer is 3D with shape $(E, 3, 16, 16)$, where $E$ is embedding size. DASH converts the 3D embedding layer to 2D of shape $(E, 768)$ by merging the remaining dimensions ($768 = 3 * 16 * 16$).
>
> When we integrate NDB into DS, we compute power $-1/2$ for the first dimension and power $-1/4$ for all other dimensions.
>
> In the table below we provide the results for these experiments. Briefly, NDB is more accurate than CN when plugged to DASH and DS. The accuracy is especially higher for DS-NDB on ImageWoof/ViT-S. Despite being more accurate than DASH, DS is 10x slower. These results show the inverse root used in the preconditioner does not impact the results and NDB method stands out even in this setting where 3D and 4D tensors are converted to 2D in the context of DASH.
>
> |dataset|model|opt|method|test acc(%)|opt step (ms)|
> |-|-|-|-|-|-|
> |nette|vit-t|DASH|CN|81.62±.65|9|
> ||||NDB|82.45±.42|9|
> |||DS|CN|81.05±1.10|104|
> ||||NDB|84.06±.55|140|
> ||
> |nette|vit-s|DASH|CN|81.44±.38|15|
> ||||NDB|83.14±.56|15|
> |||DS|CN|81.43±.38|116|
> ||||NDB|84.12±.49|145|
> ||
> |woof|vit-t|DASH|CN|66.86±.19|9|
> ||||NDB|69.86±.28|9|
> |||DS|CN|65.95±.85|105|
> ||||NDB|72.90±.11|140|
> ||
> |woof|vit-s|DASH|CN|64.14±1.02|15|
> ||||NDB|70.28±.12|15|
> |||DS|CN|63.80±.72|115|
> ||||NDB|72.08±.44|145|
>
> **References:**
>
> [1] Vineet Gupta,Tomer Koren,Yoram Singer, **Shampoo: Preconditioned Stochastic Tensor Optimization**, https://arxiv.org/pdf/1802.09568
>
> **Limitation 2**
>
> During the preparation of our work we ran 3 seeds only for a few experiments we reported in Table 1 because the runs are quite stable. We report the exact results we obtained for each seed below:
>
> |B|opt|method|freq|norm|seed|val PPL|avg PPL|std PPL|
> |-|-|-|-|-|-|-|-|-|
> |1K|DS|NDB|1|FRO|42|11.780|||
> ||||||666|11.743|11.764|.019|
> ||||||2408|11.769|||
> ||
> |1K|DASH|NDB|1|FRO|666|11.700|11.692|.011|
> ||||||2408|11.684|||
> ||
> |||||PIM|42|11.726|||
> ||||||666|11.726|11.726|.0005|
> ||||||2408|11.725|||
> ||
> |2K|DASH|NDB|1|PIM|42|11.772|||
> ||||||666|11.773|11.771|.002|
> ||||||2408|11.768|||
>
> **Limitation 3**
>
> For small models, we can afford training on a single GPU and there will be no overhead because our implementation can stack all layers in a single 3D block and perform only one call to NDB.
>
> In our answer for Limitation 2 we show results on ViT-Tiny (5M params) and ViT-Small (21M params) and we can see the running time is quite low.
>
> About larger models: our academic compute budget allows running at most 1B models, which we believe is large enough to show our method works. We will provide some results on models with 8B parameters after implementing a FSDP-compatible version of DASH.

---

> > ### Author Rebuttal · Reviewer_H8GD · 2026-04-02
> >
> > The authors have convincingly addressed my main questions and concerns!

---

> > > ### Author Response · Authors · 2026-04-02
> > >
> > > Dear Reviewer H8GD,
> > >
> > > Thank you for your response, and we are very glad that we managed to address all your main questions and concerns!
> > >
> > > A quick note: We could not help but notice that your score still appears as 3 (Weak Reject) in the interface. We were wondering if you may consider updating it to reflect the discussion.
> > >
> > > Best regards,\
> > > The authors

---

### Official Review · Reviewer_9Xeh · 2026-03-13

**Soundness:** 3
**Presentation:** 4
**Significance:** 3
**Originality:** 3
**Overall Recommendation:** 5
**Confidence:** 2

**Summary:**

The paper presents a fast but accurate second order optimizing technique, DASH. Second order optimizers are known for its high computational overhead for matrix inversion, and DASH takes a Newton-DB approach to solve the inverse root problem. Further the blocking strategy used in distributed Shampoo is adapted in favor to DASH to speed the optimization process. Overall, DASH achieves a significant improvement in GPU utilization.

**Compliance With Llm Reviewing Policy:**

Affirmed.

**Final Justification:**

The authors fully addressed the comments brought up in the original review and I will maintain my score.

**Key Questions For Authors:**

1. For evaluation on multiple optimizers, what are the main metrics used other than validation perplexity?

2. In a stylized setup, is it possible to provide some guarantees of DASH? Would this be a useful result? Are there previous works where the competing optimizers are analyzed theoretically in stylized settings?

**Limitations:**

Yes

**Strengths And Weaknesses:**

**Strengths**

- Clear presentation: The problem statement and motivation was straightforward. Limitations of the previous methods were first pointed out very clearly. The structure and flow of the paper is great.

- Theoretical insights: The proposed fixes in this paper are backed with theoretical insight that validate the motivation and performance of the proposed optimizer.


**Weaknesses**

- Extensive experiments: For the experiments ran (Llama), there could have been other metrics to evaluate the accuracy of the optimized, for instance the text output of these models compared to those trained by previous optimizers.

---

> ### Author Rebuttal · Authors · 2026-03-31
>
> We would like to thank the reviewer for their feedback. We address your concerns below.
>
> **Question 1**
>
> DASH is designed to be a faster version of Shampoo by reducing the running time of the optimizer step. With this goal in mind, we aim to preserve or improve the validation perplexity compared to the Distributed Shampoo baseline. The results in Table 1 in our paper shows we achieved these two goals, enabled by stacking the preconditioner blocks in conjunction with using the NewtonDB method and more accurate approximation of Power-Iteration run in half precision
>
> **Question 2**
>
> DASH has exactly the same guarantees as Distributed Shampoo, which is analyzed in the context of Online Convex Optimization (OCO) using the regret minimization framework. The NewtonDB and Chebyshev polynomial approaches have their own theoretical guarantees that have been widely studied long before (check Nigham, 2008 reference in our paper).
>
> One interesting direction would be to study the convergence of Distributed Shampoo optimizer when the inverse roots are approximated using NewtonDB or Coupled Newton. However, this direction is orthogonal to the purpose of our paper and we implicitly rely on the existing analysis in the OCO framework.
>
> Our results can serve as a good motivation to pursue this direction because our experiments suggest that exact root inverses are not mandatory.

---

> > ### Author Rebuttal · Reviewer_9Xeh · 2026-04-03
> >
> > The authors addressed all my main questions. I will maintain my score 5 (accept).

---

### Official Review · Reviewer_qNGa · 2026-03-13

**Soundness:** 3
**Presentation:** 2
**Significance:** 3
**Originality:** 3
**Overall Recommendation:** 4
**Confidence:** 3

**Summary:**

The authors propose the use of Newton-DB iteration and the Chebyshev polynomial approximations as more efficient approaches for computing the inverse matrix roots. They also support various optimizations like support for lower precision and reducing memory optimizations.

**Compliance With Llm Reviewing Policy:**

Affirmed.

**Final Justification:**

The authors addressed my main concerns. Thus, I am raising my score.

**Key Questions For Authors:**

1. Could the author stress the difficulties one would encounter when applying Newton-DB iteration and the Chebyshev, so that this work is not a straight application of the algorithm?
2. The authors mention in many places distributed shampoo implement some operations sequentially. Are there any fundamental reasons in the algorithm resulting in this? Is it true for other existing implementations? What difficulties need to be addressed in order to apply stacking? Otherwise, stacking feels slightly straightforward, and many of the contribution feels like engineering.

**Limitations:**

Yes

**Strengths And Weaknesses:**

Strengths:
1. This is one of the first in-depth analysis for the implementation of shampoo, and its impact can be extended to any algorithm that uses matrix inverse square root, which covers a wide range of preconditioning optimizers.

Weakness:
1. Newton-DB iteration and the Chebyshev polynomial are existing algorithms.
2. The other contributions are scattered in the paper and in general seems more like engineering effort.

---

> ### Author Rebuttal · Authors · 2026-03-31
>
> We would like to thank the reviewer for their feedback. We address your concerns below.
>
> **Answer for Question 1 & Question 2**
>
> The methods to compute inverse roots implemented in Distributed Shampoo (DS) are Eigen-Value Decomposition (EVD) and Coupled-Newton (CN). Since EVD is slow, CN was introduced to alleviate this issue because in theory it is GPU friendly, as it only requires matrix multiplications to compute inverse roots. Despite this theoretical speed-up, EVD is still the default method in DS.
>
> In our experiments, running CN in with default settings (100 steps) in DS leads to numerical errors (some values in the inverse become NaNs during the CN iteration), suggesting that CN has not been extensively tested when it was introduced. Therefore, the DS optimizer remains slow.
>
> **Summary of Contributions**
>
> In our work, we provide a solution to this problem and we categorize it into two major contributions:
> 1) **engineering contribution**, based on the observation that gradient and preconditioner blocks have the same shape and therefore can be stacked into a 3D structure to maximize the GPU utilization when using iterative approaches. Moreover, even the EVD supports batched inputs and the stacking approach slightly reduces the overall running time of the optimizer step.
>
> 2) **research contribution**, based on introducing an alternative iterative procedure to CN called NewtonDB (NDB) and explaining why NDB is better than CN by analyzing the behavior of both methods, presented in Section 3.4 in our paper.
>
> **Difficulties Encountered in CN and NDB**
>
> In DS, the input matrix $A$ for CN is scaled by Frobenius norm to make sure the convergence criteria $\lambda_{max} < 1$ is met. However, the gap between $\lambda_{max}$ and $||A||_F$ is around $10 \times$ and therefore the largest eigen-value will be 0.1 instead of 1.
>
> Our solution to this issue is an efficient Power-Iteration run in half precision applied to the stacked blocks, as explained in Section 3.5. Figure 2 shows this greatly benefits NDB.
>
> **About Chebyshev polynomial**
>
> The main challenge for the Chebyshev polynomial method was to determine the polynomial degree $d$ such that it approximates the function $f(x) = x^{-1/4}$ well for small inputs $x$. We determined a valid mapping interval for the eigen-values to be $[10^{-10}, 1+10^{-10}]$, but there are still large errors around zero, which is an important detail in the context of Shampoo.
>
> We would need the polynomial degree $d \approx 100$ to fit $f(x)$ well, which would make this method more expensive than CN or NDB. We believe this is still a good contribution to our field and hope it would be useful for the community in different contexts.
>
> **Sequential Operations in DS**
>
> We have a few explanations for this matter:
>
> 1) DS is implicitly slow when we use EVD. However, at a careful inspection of the code-base, this slow runtime is amplified by inverting the blocks in a for-loop.
>
> 2) We would like to emphasize that the repository supports a wide range of optimizers like Adam, sign descent, spectral descent (Muon) and Shampoo that combine with grafting. Because of the high degree of flexibility implemented in the repository through a lot of engineering effort, the authors might have overlooked the efficiency part, which is exactly what we address in our work.
>
> 3) By default, many PyTorch implementations update layers sequentially, which is what DS also does. However, since preconditioner blocks have the same shape for all layers, we can stack all blocks from all layers assigned to a GPU, thus having one call to the root inverse method for the 1D and 2D blocks respectively. This way, we replace the sequential calls performed in a for-loop with a single call with stacked blocks.
>
> The layer-wise approach (which is also supported by DASH) is more similar to the horizontal fusing via foreach, which all popular optimizers support, such as SGD and AdamW. However, we are not aware of any other optimizer that processes all layers assigned to one GPU in one shot as is the case for the GPU-global version of DASH (stacking the blocks from multiple layers into a single 3D structure).
>
> We would like to emphasize that even if the general idea of block stacking may sound obvious and simple, it has not been done before in the previous 3 iterations of the optimizer. The fact that we have observed the problem and provided explanations and a solution to it is already an important contribution to the research community.
>
> Making these techniques scalable at the core of the optimizer is not trivial for deep learning, as one requires not only engineering effort to implement these algorithms efficiently at scale, but also numerical analysis expertise to understand the dynamics of the methods under the hood and this is what our paper is about.
>
> To the best of our knowledge, there is no prior work that investigates the behavior of these numerical procedures in the context of stochastic optimization for deep learning.

---

> > ### Author Rebuttal · Reviewer_qNGa · 2026-04-03
> >
> > The authors addressed my main concerns.

---

### Decision · Program_Chairs · 2026-04-30

**Decision:**

Accept (regular)

**Comment:**

This paper presents DASH, a faster implementation of Distributed Shampoo. The work is well-motivated, clearly presented, and backed by theoretical insights that validate the proposed fixes. The experimental methodology is rigorous, and the practical impact is significant: DASH makes second-order optimization viable at scale without modifying model architectures or training pipelines. The main weakness raised by reviewers is the limited experimental scope — all evaluations are conducted on C4, and results on additional domains such as vision, graph, or speech would substantially strengthen the generalizability claims. The reviewers nonetheless agree the contribution is valuable, and I recommend acceptance.